# Contribution of Various Shale Components to Pore System: Insights from Attributes Analysis

Lingling Xu [1,2], Renfang Pan [1,2,*], Haiyan Hu [3] and Jianghui Meng [2,4]

1    School of Geosciences, Yangtze University, Wuhan 430100, China; 202073035@yangtzeu.edu.cn
2    Key Laboratory of Exploration Technologies for Oil and Gas Resources, Ministry of Education, Yangtze University, Wuhan 430100, China; mjh@yangtzeu.edu.cn
3    Hubei Key Laboratory of Petroleum Geochemistry and Environment, Yangtze University, Wuhan 430100, China; hyhucom@163.com
4    Cooperative Innovation Center of Unconventional Oil and Gas, Yangtze University, Wuhan 430100, China
*    Correspondence: pan@yangtzeu.edu.cn

**Abstract:** Shale pore systems are the result of the geological evolution of different matrix assemblages, and the composition of gas shale is considered to affect the pore systems in shale reservoirs. This study aimed to investigate the impact of both organic and inorganic constituents on the shale pore system, including specific surface area (SSA) and pore volume in Wufeng–Longmaxi Shale. Multiple linear regression (MLR) was employed to examine the contributions of different components to shale pore structure. The pore structure parameters, including pore SSA and pore volume, were obtained by gas adsorption experiments in 32 Wufeng–Longmaxi Shale (Late Ordovician–Early Silurian) samples. Both pore SSA and pore volume were calculated by the density functional theory (DFT) model on shale samples, and the pore types were determined by high-resolution field emission scanning electron microscopy (FE-SEM). The results of the X-ray diffractometer (XRD) analysis indicate that the Wufeng–Longmaxi Shale is dominated by quartz, clays, carbonates, feldspar, pyrite, and organic matter. Four models were made using SPSS software, all of which showed significant correlation between shale pore size and organic matter (OM) and clays. The content of organic matter played the biggest role in determining the size and structure of the pores. Although the content of quartz is the highest and serves as a rigid skeleton in shale reservoirs, it has complicated effects on the pore structure. In this study, most of the quartz is biogenetic and part of it is transformed from clays in deep shale. Therefore, these two parts of quartz are, respectively, related to organic matter and clays. In essence, the pores related to these two parts of quartz should be attributed to organic matter and clays, which also support the conclusion of the MLR models.

**Keywords:** pore structure; multiple linear regression; shale composition; diagenesis; overmature shale; attributes analysis

## 1. Introduction

The breakthrough improvement of shale gas development efficiency [1,2] has inspired the research on shale reservoirs [3–12]. In fact, a large number of pores from nanometers to micrometers are developed in shale reservoirs [7,13,14], which can provide storage space for shale gas, control the behavior of fluid in the pores [5,15,16], and ultimately determine the productivity potential of shale reservoirs. Therefore, the pore system is at the heart of our understanding of shale reservoirs. Dominant factors that control the shale pore systems have long been a question of great interest in a wide range of shale gas fields [12,17–19]. However, the shale pore systems, which are directly carried by the minerals matrix and organic matter (abbreviated as "OM") [4,20], were formed after sedimentary evolution and were controlled by many geological factors, such as shale mineralogy, OM (including abundance and type), burial depth, diagenesis, maturity, and preserved condition [12,19–21]. The constituents of shale consist of two parts: inorganic minerals and OM. Quartz, feldspar,

carbonates, and pyrite are generally considered as brittle minerals, while clays and OM as ductile components [22]. It has been demonstrated by many researchers that OM-hosted pores are strongly associated with the OM maturity [5,18,20,23], and the mineral matrix pores are mainly related to burial depth and diagenesis [24]. Recently, there has been renewed interest in the contribution of shale composition to the pore systems [12,25,26], which highly demonstrated the importance of quartz to the shale pore network. Some research reported that the pore specific surface area (abbreviated as "SSA") and pore volume of gas shale increases with the increasing abundance of quartz [13,27–30], and this positive correlation was interpreted to be caused by these rigid minerals that inhibit pore collapse, but other research suggests that quartz does not significantly contribute to the pore network of shales [15,17]. The same phenomenon applies to the study between other shale components and pore structure parameters [21,26,31], which will be well discussed in Section 4. Even though it has been studied in many publications [17,32,33], the effects of shale composition (including organic and inorganic constituents) on pore system are not fully understood. The existing studies fail to resolve the collinearity between independent variables [14,30,34]. So far, only a few studies have noted the collinearity relationship between the content of quartz and OM in the research of the effect of shale components on pore network [13,29,35]. In other words, the highly positive correlation between the content of quartz and shale pore structure parameters may be triggered by high TOC rather than the quartz. Moreover, previous studies often directly use correlation coefficients to represent the degree of correlation between variables when judging the correlation between factors. When the number of samples is small, the correlation coefficient fluctuates greatly and more easily approaches 1 [28,36]. Contrary to previous work, this study introduces test probability P to determine the possibility of correlation between variables using SPSS version 29.0, so as to ensure that the results are statistically significant. In the light of recent research, it is becoming extremely difficult to ignore the contributions of shale composition in shale pore network towards overall and effective porosity. This is particularly relevant in the light of the discovery of various type pores [4,37,38]: To what degree are these pore SSA and pore volume provided by minerals versus OM in the overmature shales? In addition, a systematic understanding of how shale composition contributes to the pore network is still lacking.

This essay investigates the contribution of various compositions to the pore system in overmature shale. The specific objective of this study was to establish MLR models between shale constituents and pore structure parameters to determine the components and contribution proportion that have significant contributions to pore structure. The MLR analysis was conducted by SPSS software to eliminate the collinear relationship between independent variables and quantify the contribution of the independent variables to the dependent variables. Consequently, this research will contribute to a deeper understanding of the pore system in overmature shale reservoirs, clarifying the degree to which different components contribute to shale pore SSA, and that pore volume is also important for shale gas exploration.

## 2. Materials and Methods

### 2.1. Samples and Background

Samples used for analysis in this paper were collected from fresh cores of the Wufeng–Longmaxi Shale in South Sichuan shale clays (China) which had experienced similar burial history with a maximum burial depth of 6000–7000 m [39]. Hence, it is approximately believed that all samples had undergone similar diagenetic evolution, and the difference in pore development is mainly caused by their heterogenous composition. Then, the main reason for the differences in pore structure characteristics of the selected samples is their various components. As the Wufeng–Longmaxi Shale is overmature, it is difficult to distinguish the kerogen type by optical observation or Rock-Eval data [40–42]. In this study, the kerogen type was identified by the stable carbon isotopic values (Table 1) according to previous research [40]. The stable carbon isotopic values of selected samples were measured using a FLASH HT EA-MAT 253 IRMS, and the values were calibrated to the PDB standard.

**Table 1.** Standard of kerogen type for the Wufeng–Longmaxi Shales based on carbon isotopic composition (adopted from [40]).

| Kerogen Types | $\delta^{13}$C (‰) |
|---|---|
| I | $<-29$ |
| II$_1$ | $-29$ to $-27$ |
| II$_2$ | $-27$ to $-25$ |
| III | $>-25$ |

The shale samples lack vitrinite particles because the Wufeng–Longmaxi Shale was deposited in the Late Ordovician to Early Silurian, and the reflectance of bitumen (abbreviated as "Rb") was measured to reflect the maturity of shales [43]. In this paper, the Ro* was calculated by the equation: Ro* = 0.938Rb + 0.3145 according to [43], and the random reflection of solid bitumen was measured by a Zeiss Axio Scope A1 reflected-light microscope with a J&M MSP 200 microscope photometer. The mineral components were measured on an X' Pert3 Powder diffractometer using K-alpha radiation (XRD). Using a scan step of 0.02° and scanning speed at 2°/min, data were obtained at room temperature in 2θ range of 5° to 45°. Quantitative analysis of minerals was calculated by the equation following the Chinese Oil and Gas Industry Standard (SY/T 5163-2018) [12]. An LECO carbon–sulfur analyzer (CS230) was utilized to obtain the TOC content of these samples. Finally, the shale composition results, including organic and inorganic constituents, were normalized to 100% (Table 2).

**Table 2.** The composition, Ro*, and $\delta^{13}$C$_{PDB}$ of kerogen of samples from Wufeng–Longmaxi Shales.

| Samples ID | Depth (m) | Kerogen $\delta^{13}$C$_{PDB}$ (‰) | Rb (%) | Ro* (%) | Quartz (%) | Clays * (%) | Carbonates ** (%) | Feldspar † (%) | Pyrite (%) | TOC (%) |
|---|---|---|---|---|---|---|---|---|---|---|
| Y101H4-4-1 | 4124.27 | −30.1 | / | / | 37.88 | 30.26 | 16.76 | 11.66 | 1.10 | 2.34 |
| Y101H4-4-2 | 4131.55 | −30.6 | 3.032 | 3.16 | 42.62 | 34.14 | 11.55 | 7.33 | 1.64 | 2.71 |
| Y101H4-4-3 | 4140.22 | −30.5 | 2.907 | 3.04 | 64.70 | 18.34 | 6.84 | 3.06 | 2.02 | 5.04 |
| Y101H4-4-4 | 4144.14 | −30.9 | 2.88 | 3.02 | 56.91 | 13.36 | 19.32 | 2.56 | 2.05 | 5.78 |
| Y101H4-4-5 | 4145.86 | / | / | / | 58.49 | 10.45 | 22.98 | 2.02 | 1.12 | 4.94 |
| Y101H4-4-6 | 4150.60 | / | 2.98 | 3.11 | 57.59 | 29.87 | 4.90 | 3.69 | 0.00 | 3.95 |
| Y101H4-4-7 | 4154.35 | / | / | / | 35.55 | 44.11 | 15.45 | 4.39 | 0.00 | 0.5 |
| Y101H2-7-1 | 4102.25 | / | / | / | 30.16 | 47.68 | 11.70 | 5.72 | 1.34 | 3.39 |
| Y101H2-7-2 | 4109.57 | / | / | / | 35.47 | 48.01 | 5.66 | 4.72 | 3.12 | 3.01 |
| Y101H2-7-3 | 4139.05 | −30.0 | 3.002 | 3.13 | 47.23 | 35.30 | 4.77 | 6.30 | 1.97 | 4.44 |
| Z206-1 | 4244.57 | −29.5 | 2.43 | 2.6 | 47.43 | 32.77 | 8.05 | 6.96 | 1.26 | 3.53 |
| Z206-2 | 4254.05 | / | / | / | 44.95 | 32.30 | 7.87 | 10.11 | 2.25 | 2.52 |
| Z206-3 | 4258.64 | / | / | / | 43.00 | 28.39 | 14.47 | 9.48 | 2.11 | 2.55 |
| Z206-4 | 4266.76 | −30.5 | 2.471 | 2.63 | 72.57 | 13.56 | 5.91 | 2.07 | 1.35 | 4.54 |
| Z206-5 | 4274.36 | −30.4 | / | / | 24.96 | 32.44 | 37.92 | 3.23 | 0.73 | 0.71 |
| L204-1 | 3823.55 | / | / | / | 34.41 | 43.78 | 10.96 | 4.92 | 2.40 | 3.54 |
| L204-2 | 3835.91 | / | / | / | 49.56 | 16.57 | 24.84 | 2.68 | 2.41 | 3.93 |
| L204-3 | 3842.21 | −30.7 | 2.698 | 2.85 | 69.49 | 8.53 | 13.90 | 1.84 | 0.68 | 5.56 |
| L204-4 | 3844.71 | / | / | / | 50.26 | 8.27 | 33.80 | 2.18 | 0.67 | 4.82 |
| L205-1 | 4012.06 | / | / | / | 38.26 | 33.89 | 13.50 | 9.81 | 1.68 | 2.86 |
| L205-2 | 4024.08 | −29.7 | 2.567 | 2.72 | 56.67 | 22.57 | 11.74 | 4.31 | 1.45 | 3.26 |
| L205-3 | 4028.30 | / | / | / | 61.48 | 7.43 | 24.01 | 1.40 | 1.00 | 4.68 |
| L205-4 | 4032.58 | / | / | / | 57.85 | 11.96 | 22.06 | 2.31 | 1.74 | 4.07 |
| L205-5 | 4034.44 | −29.4 | 2.814 | 2.95 | 51.31 | 15.18 | 22.98 | 4.27 | 2.75 | 3.51 |
| N213-1 | 2575.28 | −30.9 | 2.48 | 2.64 | 41.67 | 7.74 | 43.99 | 1.47 | 1.51 | 3.61 |
| N216-1 | 2315.30 | / | / | / | 38.98 | 9.66 | 36.00 | 1.53 | 10.40 | 3.42 |
| N216-2 | 2320.68 | −30.4 | 2.65 | 2.8 | 46.67 | 7.76 | 38.36 | 2.83 | 0.90 | 3.49 |
| W202-1 | 2560.20 | −29.2 | 2.53 | 2.69 | 76.37 | 3.89 | 10.82 | 0.67 | 0.37 | 7.88 |
| W202-2 | 2573.20 | / | / | / | 42.06 | 2.64 | 50.57 | 1.46 | 0.54 | 2.73 |

The shale composition results, including organic and inorganic constituents, are given in mass percent normalized to 100% (wt.%). Depth here refers to the depth of the core where the sample is located. TOC = total organic carbon content, wt.%; Ro* = equivalent vitrinite reflectance calculated from the reflectance of bitumen (Rb), Ro* = 0.938Rb + 0.3145 [43]; * Clays = illite + illite/smectite + chlorite; ** Carbonates = calcite + dolomite; † Feldspars = orthoclase + plagioclase; / = samples with no available data.

### 2.2. FE-SEM

In order to focus on the development of pores related to a certain component and their types depending on their host sites in shales, 28 samples with a relatively high content of a certain component (quartz, clays, carbonates, feldspar, pyrite, TOC) were selected for FE-SEM observation. All samples were prepared by Ar-ion-milling for a much flatter surface [3] before using FE-SEM. The detailed aspects of the FE-SEM observation have been adequately described by previous authors and we recommend readers to [3].

### 2.3. Low-Pressure Gas Adsorption

Low-pressure gas adsorption is an effective approach for measuring the pore SSA and pore volume of shales [5–8,44], which can effectively reflect capacity of shale gas and heterogeneity of shales [27,45,46].

Before gas adsorption experiments, all samples were ground into about 35 mesh and then degassed under vacuum for 10 h at 120 °C to remove volatiles. The experiments were conducted by a Quantachrome® Autosorb iQ instrument. $N_2$ physisorption experiments were performed at −195.8 °C. More detailed explanation of the $N_2$ adsorption experiment is presented in previous studies [8,44]. $CO_2$ adsorption experiments were carried out at 0 °C with the same instrument. Detailed information about the $CO_2$ adsorption experiments can be found in previous studies [47,48].

The pore structure was represented by pore SSA and pore volume in this paper. Both the pore SSA and pore volume were calculated by the density functional theory (abbreviated as "DFT") from the adsorption isotherms since DFT models cover the range of micro- to macropores, fit different adsorbates ($N_2$ and $CO_2$), and take advantage of reflecting various pore morphologies [44,46,49,50]. More detailed information and explanation of this theory can be found in previous studies [48,49].

### 2.4. Multiple Linear Regression

MLR analysis is a statistical approach to investigate the interdependent quantitative relationship between two or more variables. In this study, MLR analysis was conducted by using IBM SPSS Statistics 29.0 automatically. In the MLR models, the contents of quartz, clays, carbonates, feldspar, pyrite, and TOC served as independent variables, and the pore SSA and pore volume obtained by $CO_2$ and $N_2$ served as dependent variables, respectively. A stepwise method was adopted to select the independent variables that contribute most to the dependent variables and meet the conditions (probability of $F \leq 0.05$) in the model and remove any remaining variables (probability of $F \geq 0.10$) that do not meet the set conditions [51]. Collinearity diagnosis was added to exclude collinearity relationship between the independent variables. In this study, the quality criterion of MLR model includes the following: (1) F-value is used to test the significance of the model (the larger the F-value, the smaller the *p*-values of significance probability, indicating that the regression model is effective); (2) adjusted $R^2$, which directly reflects the proportion of the regression value in the model (the closer the adjusted $R^2$ is to 1, the better the quality of the regression); (3) T-values test the significance of regression parameter for each independent variable (a large T-values implies a small *p*-value, which means that the independent variable contribute significantly to the equation).

## 3. Results

### 3.1. Components, Ro*, and Kerogen Type

The Ro* and $\delta^{13}C_{PDB}$ for the kerogen from all the samples are listed in Table 2, and the main minerals developed in the Wufeng–Longmaxi Shale are summarized in a boxplot (Figure 1). The Wufeng–Longmaxi Shale is mainly composed of variable amounts of quartz, clays, carbonates (calcite and dolomite), feldspar, and TOC, with a small amount of pyrite (Table 2; Figure 1). The inorganic constituents show an inverse relationship in terms of the proportion, that is, the increase of one mineral is compensated by a decrease in the remaining minerals. The content of quartz ranges between 25.14% and 82.91%, with an

average of 50.82% (Table 2; Figure 1). The content of clays varies between 1.52% and 49.50%, with an average of 22.96%. The content of carbonates ranges between 4.99% and 51.99%, with an average of 19.72%. The content of feldspar is relatively small, ranging from 0.72% to 11.94%, with a mean value of 4.42%. The content of pyrite is smallest, with an average of 1.81%, the content of all samples, except one sample with 10.77% content, does not exceed 3.2%. The content of TOC ranges between 0.50% to 7.88%, with a mean value of 3.70% (Table 2). In particular, the content of TOC is positively correlated with quartz content, but negatively correlated with clays content. The Ro* ranges from 2.60% to 3.16%, with mean Ro* of 2.88% (Table 2), indicating that all these shale samples are overmature. The Wufeng–Longmaxi Shales have $\delta^{13}C$ values of less than −29‰ (Table 2), and are type I [40].

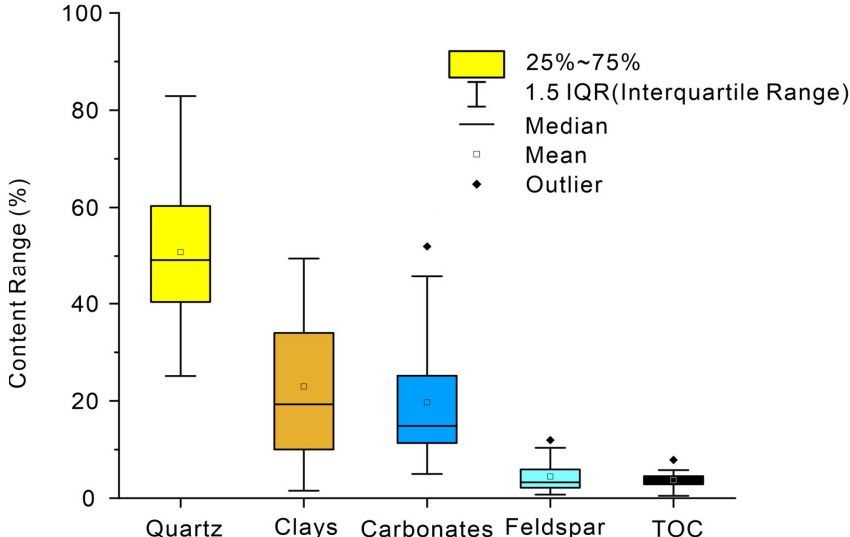

**Figure 1.** Boxplot for the main composition of the Wufeng–Longmaxi Shale.

### 3.2. Pores Related to Various Components

Different components of shale develop various types of pores (terminology after [4]). Pores related to quartz include interparticle (abbreviated as "interP") pores (Figure 2a,b) and intraparticle (abbreviated as "intraP") pores (Figure 2c,d), among which the intraP pores are rare; only a very small number of them developed in a few individual grains (Figure 2a,b), with more interP pores developing between quartz individuals or quartz and other minerals such as feldspar, carbonates, or illite (Figure 3b). The number of pores associated with clays is relatively large, and there are a large number of cracks that developed between clays or within them (Figures 3 and 4). A large number of irregular pores developed between clays and other brittle minerals (Figure 3b). However, the degree of pore development in argillaceous shale without OM is extremely low (Figure 4), which only developed few cracks in it. Carbonates developed more intraP pores than quartz grains from the FE-SEM images (Figure 5), which are suspected to be formed from the acid dissolution by circulating acid fluid [6,35]. InterP pores formed at the edge of the grains when along contact with other grains (Figure 5a,b). Feldspar also shows development of developed interP pores and interP pores (Figure 6c,d). Pyrite is the least abundant compared to other components described above in all samples. There are two types of pores related to pyrite, one is intercrystalline pores within pyrite framboids (Figure 6a), and the other is interP pores between pyrites (Figure 6b), which is often associated with OM. Pores associated with OM are the most abundant pores that can be observed in FE-SEM images. At a large scale, OM is dispersed in matrix minerals (Figure 7a,d) and these are connected with each other through the edges of mineral particles. There are nano- to micropores (Figure 7b) that developed within OM. After local amplification, it can be found that there are numerous nanopores that developed within OM, similar to a honeycomb (Figure 7c).

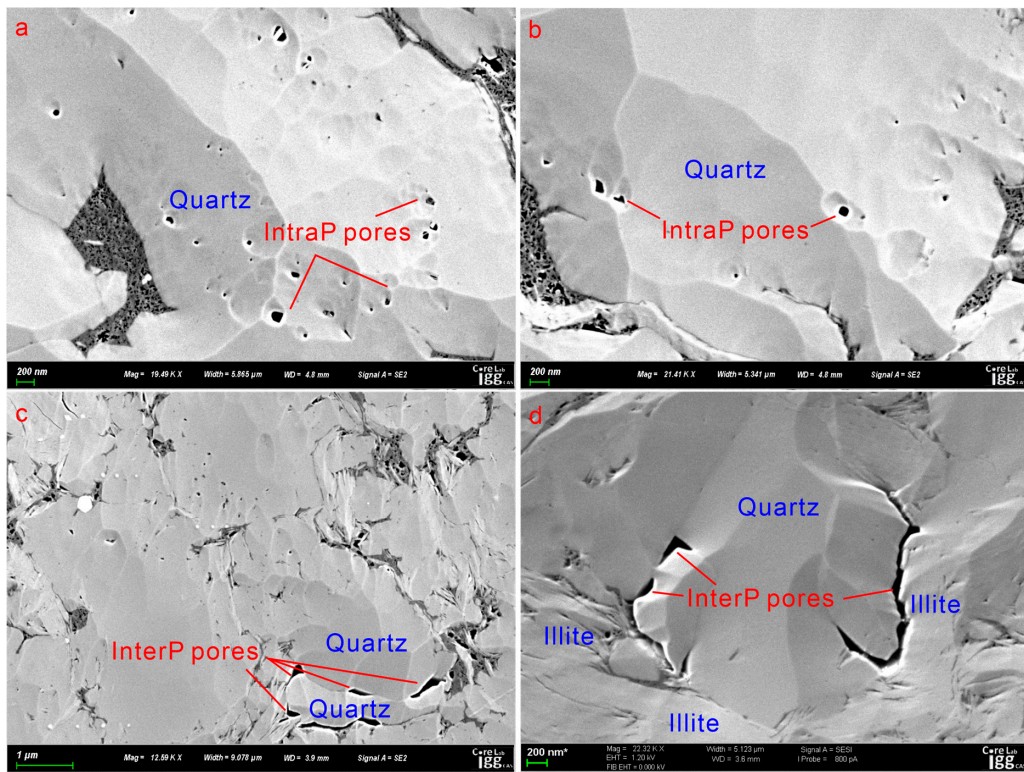

**Figure 2.** Pores associated with quartz: (**a**,**b**) intraP pores developed within quartz grain; (**c**) interP pores developed between quartz grains; (**d**) interP pores developed between quartz and illite [52].

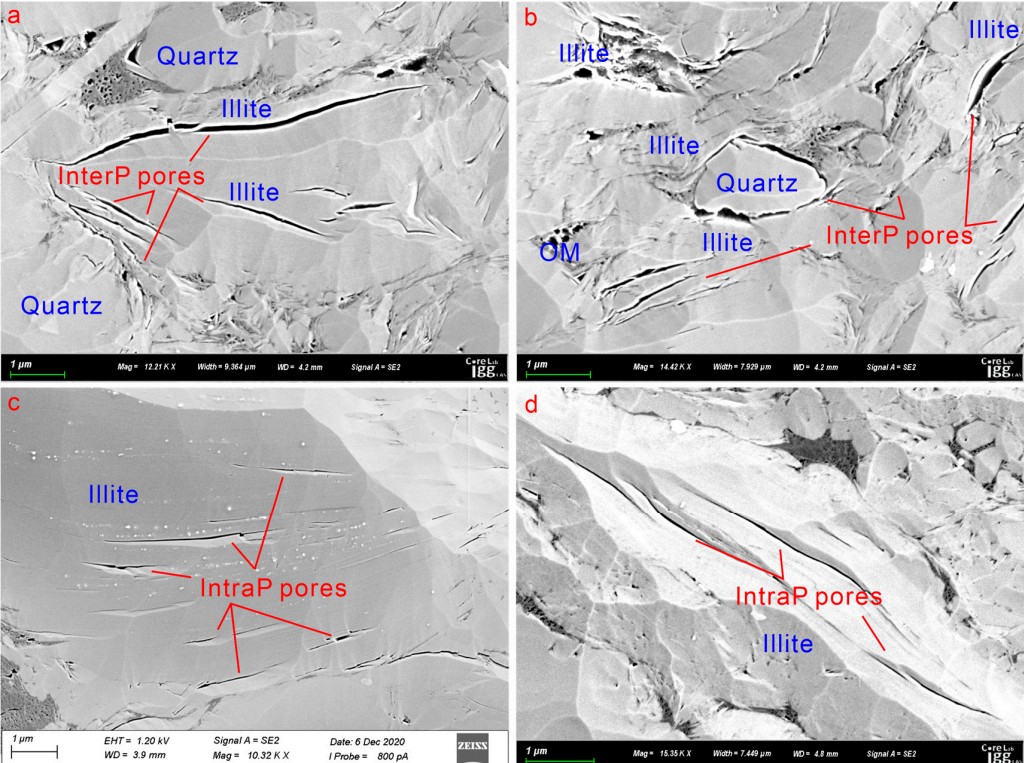

**Figure 3.** Pores related to clays, presenting as a set of parallel to subparallel cracks/slits generally running horizontally: (**a**,**b**) interP pores developed between clays and other grains; (**c**,**d**) intraP pores developed within clays.

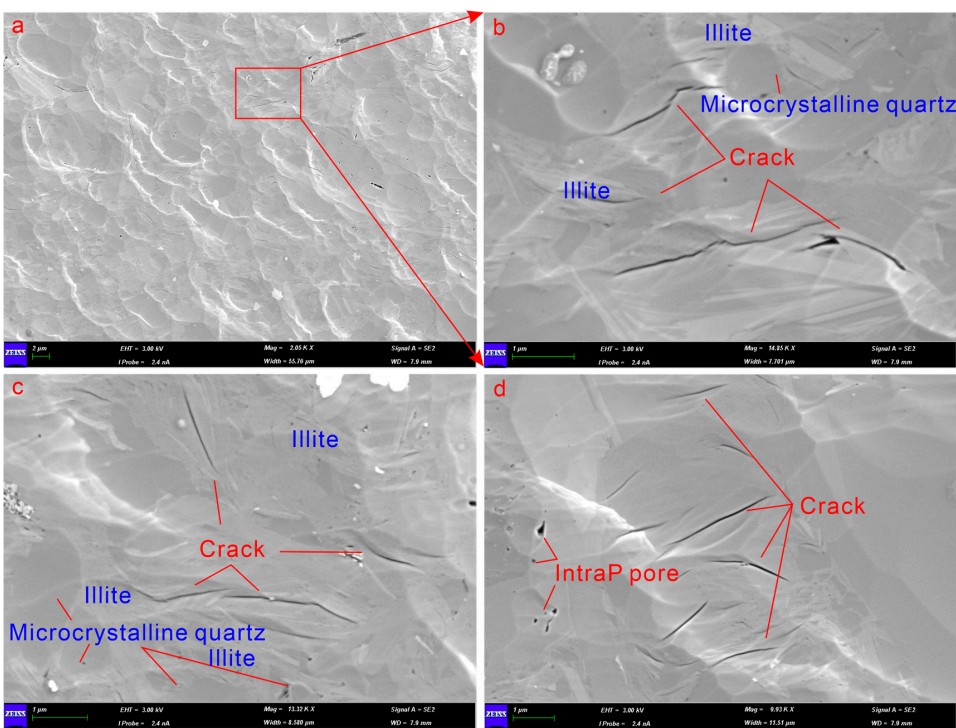

**Figure 4.** FE-SEM images of argillaceous shale, sample Y101H4-4-7, with 0.5% TOC, which developed few inorganic pores and cracks. (**a**) FE-SEM image of sample Y101H4-4-7 on a large scale; (**b**) partial enlarged image in (**a**); (**c**) cracks and microcrystalline quartz observed in the FE-SEM image of sample Y101H4-4-7; (**d**) cracks and intrP pores developed in the argillaceous shale.

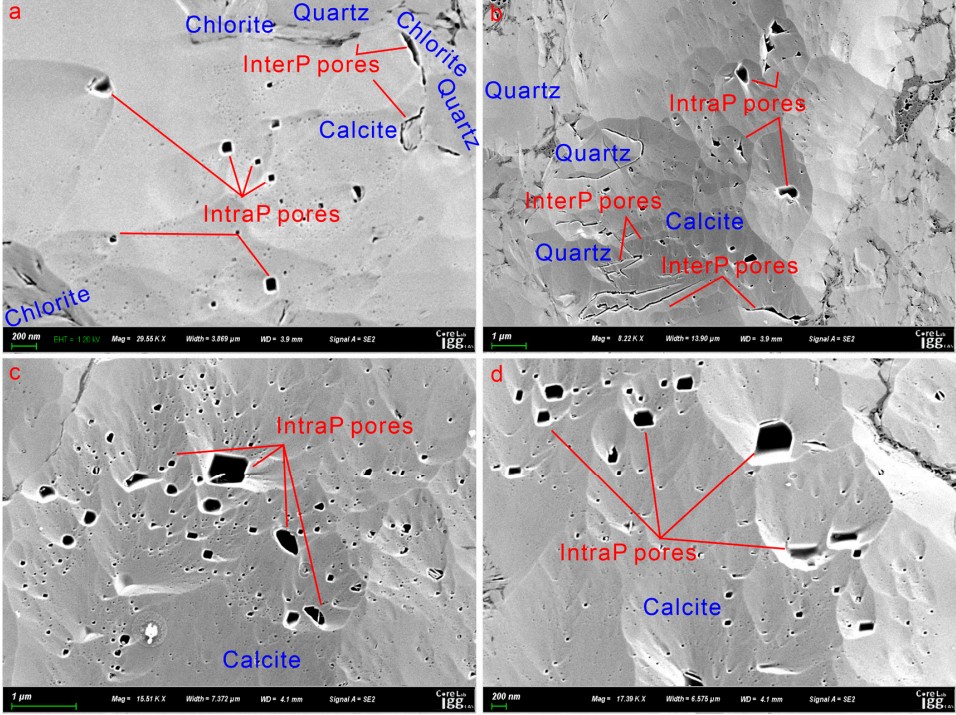

**Figure 5.** Pores related to carbonates. (**a**) sample ID = Y101H4-4-2, intraP pores developed in calcite particle and interP pores developed between calcite and other grains (quartz or chlorite); (**b**) sample ID = Y101H4-4-3, intraP pores developed in calcite particle and interP pores developed between calcite and quartz; (**c**) sample ID = L204-4, intraP pores developed in calcite; (**d**) sample ID = L204-4, intraP pores developed in calcite.

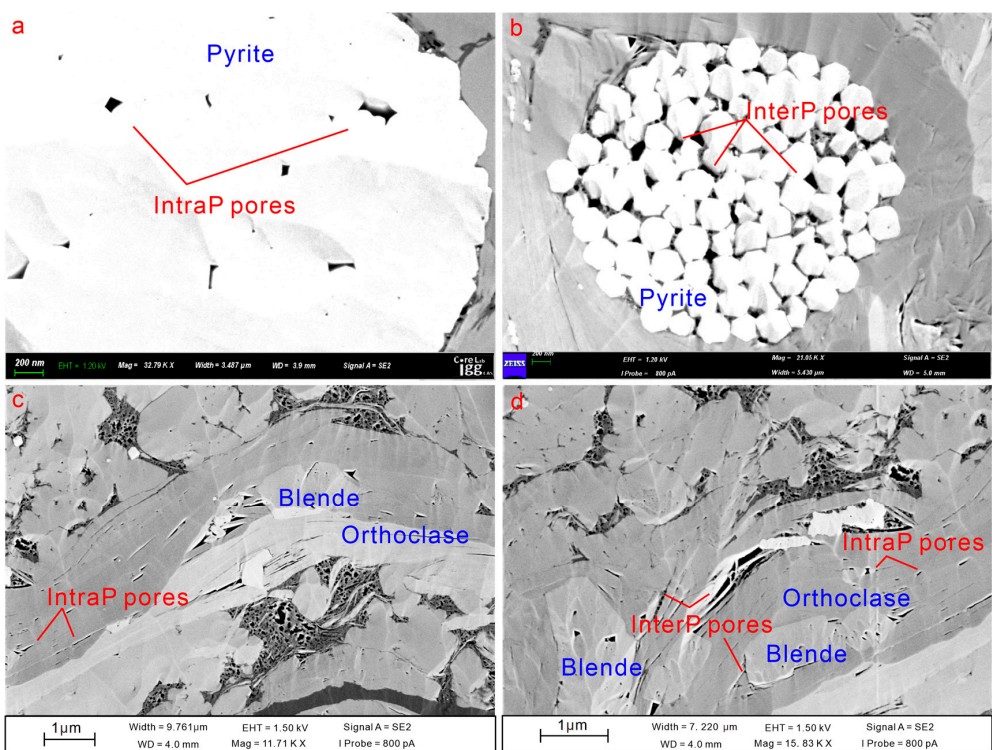

**Figure 6.** Pores related to pyrite and feldspar: (**a**,**b**) intraP pores developed within pyrite or between pyrite grains; (**c**) intraP pores developed within orthoclase grain; (**d**) interP pores developed between orthoclase and other grains.

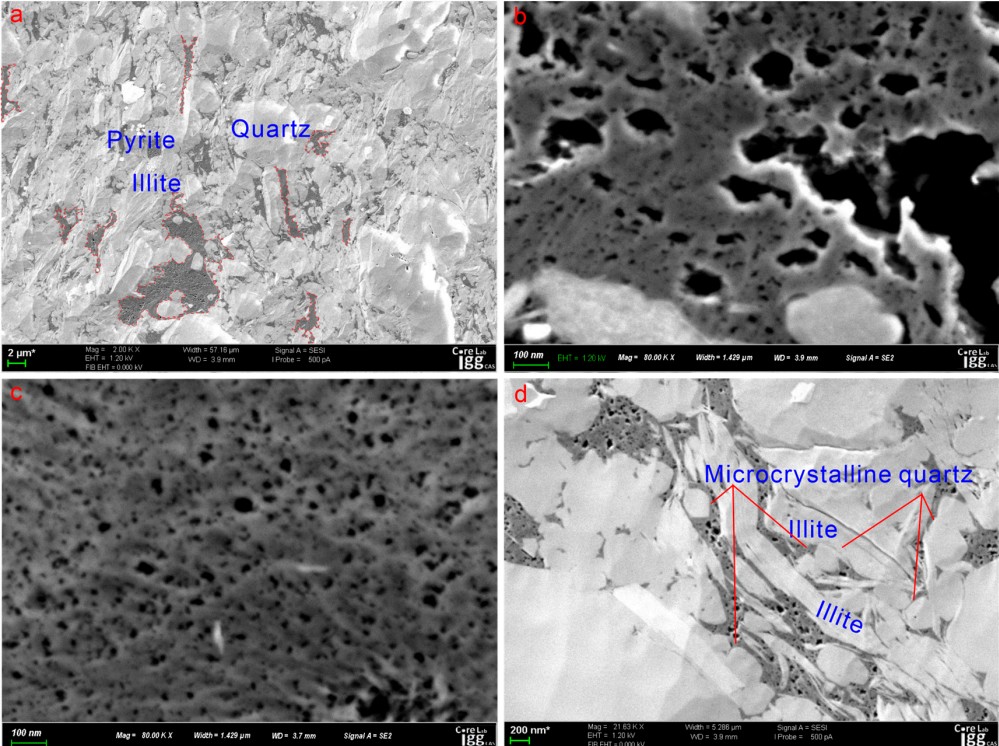

**Figure 7.** OM-hosted pores: (**a**) sample ID = Y101H2-7-3, OM is dispersed in matrix minerals at a large scale; (**b**) sample ID = Y101H4-4-2, nano- to micropores developed within OM; (**c**) sample ID = Y101H4-4-1, honeycomb-like OM-hosted pores; (**d**) sample ID = Y101H2-7-3, OM-hosted pores developed within clays.

### 3.3. Pore SSA and Pore Volume

Low-pressure $CO_2$ adsorption isotherms are shown in Figure 8. The pore SSA and volume obtained by low-pressure $CO_2$ were calculated through the DFT (Table 3). The $CO_2$ micropore volumes have the smallest variation range, from 0.003 $cm^3/g$ to 0.009 $cm^3/g$, with a mean value of 0.006 $cm^3/g$. The $CO_2$ SSA ranges from 9.392 $m^2/g$ to 28.227 $m^2/g$, with an average of 18.903 $m^2/g$. The low-pressure $N_2$ adsorption isotherms are shown in Figure 9. The pore SSA and volume obtained by low-pressure $N_2$ were calculated by the DFT (Table 3). The $N_2$ SSA varies from 6.14 $m^2/g$ to 34.48 $m^2/g$, with a mean value of 22.31 $m^2/g$, and the $N_2$ volume ranges between 17.64 $cm^3/g$ and 38.56 $cm^3/g$, with an average of 27.01 $cm^3/g$.

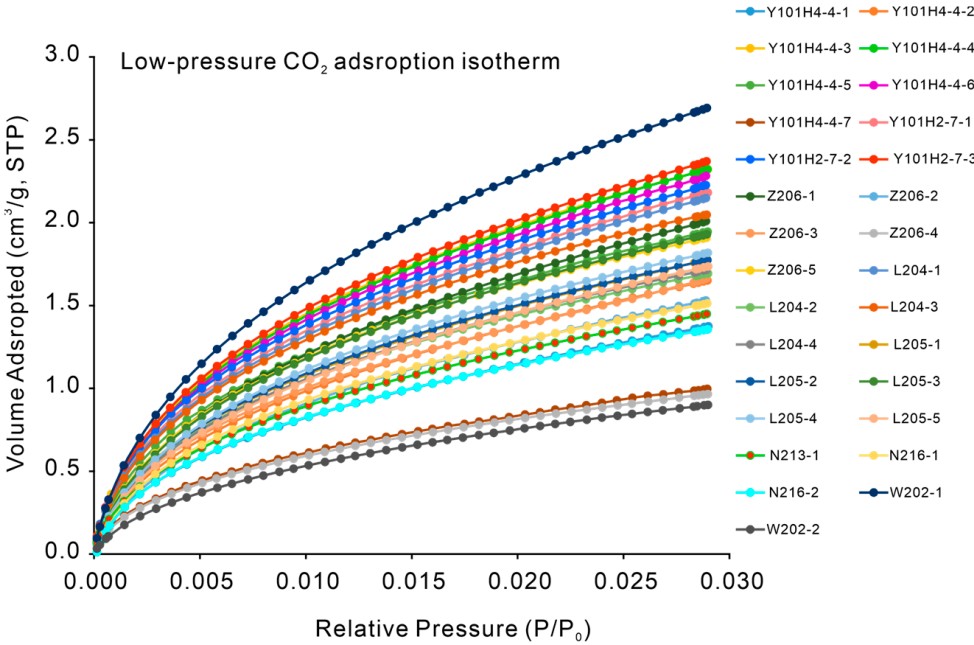

**Figure 8.** Low-pressure $CO_2$ adsorption isotherms of the 29 samples from the Wufeng–Longmaxi Shales.

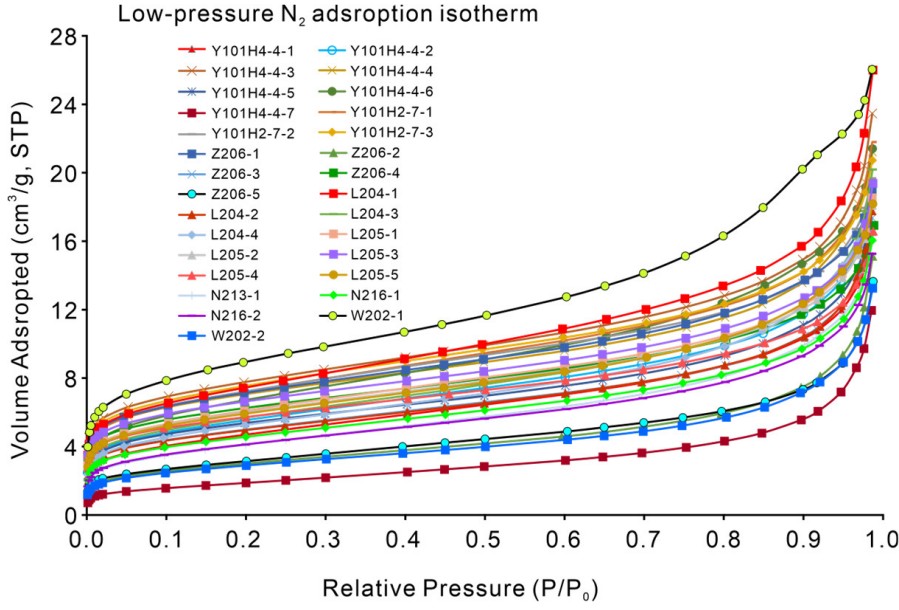

**Figure 9.** Low-pressure $N_2$ adsorption isotherms of the 29 samples from the Wufeng–Longmaxi Shales.

**Table 3.** Pore parameters of samples from Wufeng–Longmaxi Shales.

| Samples ID | $CO_2$ | | $N_2$ | |
|---|---|---|---|---|
| | SSA $(m^2/g)$ | V $(\times 10^{-3} cm^3/g)$ | SSA $(m^2/g)$ | V $(\times 10^{-3} cm^3/g)$ |
| Y101H4-4-1 | 14.406 | 4 | 17.26 | 24.2 |
| Y101H4-4-2 | 17.392 | 5 | 21.76 | 28.36 |
| Y101H4-4-3 | 24.255 | 7 | 32.87 | 34.37 |
| Y101H4-4-4 | 24.371 | 7 | 30.28 | 31.28 |
| Y101H4-4-5 | 20.314 | 6 | 21.93 | 26.86 |
| Y101H4-4-6 | 23.936 | 7 | 25.69 | 31.25 |
| Y101H4-4-7 | 10.342 | 3 | 6.14 | 17.64 |
| Y101H2-7-1 | 22.985 | 7 | 29.37 | 31.93 |
| Y101H2-7-2 | 23.251 | 7 | 30.04 | 29.12 |
| Y101H2-7-3 | 24.726 | 7 | 31.69 | 30.43 |
| Z206-1 | 21.098 | 7 | 30.08 | 27.94 |
| Z206-2 | 16.067 | 5 | 10.77 | 21.69 |
| Z206-3 | 17.561 | 5 | 20.5 | 25.35 |
| Z206-4 | 20.049 | 6 | 26.8 | 24.39 |
| Z206-5 | 10.072 | 3 | 11.76 | 19.37 |
| L204-1 | 22.495 | 7 | 30.01 | 37.71 |
| L204-2 | 17.719 | 5 | 19.53 | 25.96 |
| L204-3 | 21.34 | 6 | 22.04 | 29.39 |
| L204-4 | 17.956 | 5 | 19.86 | 27.21 |
| L205-1 | 18.518 | 6 | 18.02 | 27.24 |
| L205-2 | 18.593 | 6 | 24.04 | 26.57 |
| L205-3 | 20.266 | 6 | 28.42 | 27.96 |
| L205-4 | 19.049 | 6 | 24.09 | 23.87 |
| L205-5 | 18.323 | 6 | 23.73 | 26.6 |
| N213-1 | 15.139 | 5 | 14.12 | 23.33 |
| N216-1 | 15.955 | 5 | 17.39 | 23.41 |
| N216-2 | 14.378 | 4 | 14.48 | 22.26 |
| W202-1 | 28.227 | 9 | 34.48 | 38.56 |
| W202-2 | 9.392 | 3 | 9.98 | 19.17 |

SSA = specific surface area; V = volume.

*3.4. Multiple Linear Regression*

Four MLR models with corresponding adjusted $R^2$, F-values, and *p*-values conducted by SPSS are shown in Table 4. The *p*-values of the four MLR models were all less than 0.001, indicating these four MLR modes were all statistically significant. The adjusted $R^2$ ranged from 0.753 to 0.931, showing that there are significant correlations between the residual independent variables and the dependent variables. The factors that contribute to the pore SSA and pore volume are the same. Because there is no statistically significant correlation between quartz, carbonates, feldspar, pyrite, and the $CO_2$ SSA, $CO_2$ volume, $N_2$ SSA, and $N_2$ volume, the independent variables quartz, carbonates, feldspar, and pyrite were excluded from the MLR models with *p*-values more than 0.1. The results show that $CO_2$ SSA, $CO_2$ volume, $N_2$ SSA, and $N_2$ volume were significantly correlated with TOC and clays (Table 4). Furthermore, there are no collinear relationships between clays and TOC in these four MLR models. TOC was the most significant contributor to the total SSA and volume of shale pores since the correlation coefficient is larger, followed by clays (Table 4). The contribution of TOC to the four parameters of shale pore structure is arranged from large to small, as follows: $CO_2$ SSA, $CO_2$ volume, $N_2$ volume, $N_2$ SSA. The situation for clay minerals is different: $N_2$ volume, $CO_2$ SSA, $CO_2$ volume, $N_2$ SSA.

**Table 4.** MLR models with corresponding R, adjusted $R^2$, and standard error (SE) of the estimate.

| Dependent Variable | Model | | | | TOC | | | Clays | | |
|---|---|---|---|---|---|---|---|---|---|---|
| | Equation | Adjusted $R^2$ | F | $p$ [a] | r | t | $p$ | r | t | $p$ |
| $CO_2$ SSA | S = 1.145T + 0.705C | 0.931 | 190.855 | <0.001 | 1.145 | 19.460 | <0.001 | 0.705 | 11.974 | <0.001 |
| $CO_2$ volume | V = 1.099T + 0.674C | 0.852 | 81.611 | <0.001 | 1.099 | 12.728 | <0.001 | 0.674 | 7.804 | <0.001 |
| $N_2$ SSA | S = 1.037T + 0.657C | 0.753 | 43.766 | <0.001 | 1.037 | 9.299 | <0.001 | 0.657 | 5.890 | <0.001 |
| $N_2$ volume | V = 1.071T + 0.708C | 0.817 | 63.452 | <0.001 | 1.071 | 11.152 | <0.001 | 0.708 | 7.364 | <0.001 |

Note that S = specific surface area, T = content of TOC, C = content of clays, F = sum of regression square/sum of residual mean square, $p$ [a] = probability of the significance for the model, r = coefficient of the independent variables, T = standardization coefficient/residual, $p$ = test probability value of independent variables.

## 4. Discussion

According to XRD analysis, the samples are mainly composed of OM, quartz, clays, carbonates, feldspar, and a small amount of pyrite. Although each constituent contributes to the total porosity (Figures 4–7), their contributions to the total pore SSA and volume vary, as indicated by the result of MLR in this paper (Table 4). The constituents of shales may be classified into brittle and ductile in terms of their mechanical behavior. The four MLR models show that the pore SSA and pore volume of the overmature shale samples are mainly contributed by ductile components rather than brittle minerals, in particular OM, which is in agreement with some previous studies [3,5,6,8,20]. In this study, we also see a positive relationship between total porosity and the contents of clays [14,20] and no correlations between total pore volume and the contents of quartz, or the contents of carbonates. The contribution of brittle minerals, clays, and OM to the pore development of overmature shales is discussed as follows, combined with the results of FE-SEM images.

### 4.1. Contribution of Brittle Minerals to Pore Development

Quartz, the most abundant component in marine shales [53], hardly contributes to the SSA and volume of shale pores, as indicated by the results of MLR (Table 4). However, many publications have reported a positive relationship between the contents of quartz and pore SSA and pore volume, and have concluded that quartz contributes to the development of shale pores [13,14,29,33]. In fact, quartz in marine shales mainly originates from three sources: detrital quartz, biogenic quartz, and clay minerals transformation [54–56]. In most organic-rich marine shales, biogenic quartz accounts for the majority [13,14,25,26,55,57]. As a result, TOC increases with the increasing content of quartz, particularly biogenic quartz [13,25,29,30,58,59]. Indeed, as [35] pointed out, the positive correlations between quartz and pore parameters are attributed to the positive correlation between quartz and OM. Our research sample contains parts of biogenic quartz, and the content of quartz is positively correlated with TOC (Figure 10a). Although we can obtain similar conclusions by using the data from one of the wells as an example and performing a linear regression analysis between the quartz content and pore structure parameters (as shown in Figure 11), these conclusions seem untenable. Maintaining primary pores or generating secondary pores are two ways for shale reservoirs to maintain high porosity [1,20,35,60]. However, on the one hand, we did not observe a significant number of pores associated with quartz in the FE-SEM images, regardless of the type of quartz (as seen in Figure 2). In Section 3.2, we identified three types of pores that are associated with quartz grains. The first type is interP pores located between quartz grains, which are detrital or biogenic quartz, whereas smaller-sized quartz particles transformed by clays typically form within the clays (Figure 12). Under FE-SEM, there are very few pores observed between quartz particles. The second type of pores related to quartz is the dissolution pore [14,57]. The SEM results show that the pores produced by a single particle due to dissolution are limited (Figure 2) compared to carbonates (Figure 5), which is consistent with the previous study [14]. The third type of pores is related to other minerals. The contact between quartz and other minerals (including OM) is more easily compressed due to differences in hardness [17]. Additionally, in this study, there are many quartzes transformed from clay minerals (Figure 12). Since most of this microcrystalline quartz with small size develops within clays [24,54,61,62], it is easy

to identify. Nevertheless, numerous pores developed between clays and microcrystalline quartz, and we suggested attributing these pores to clays rather than quartz. Essentially, the higher the content of clays, the more pore spaces are formed in clays. Moreover, the pores related to biogenetic quartz are also linked to the content of OM. Overall, the number of pores contributed by quartz itself is very limited [15]. Although the number of secondary pores generated by quartz is limited, they play a key role in maintaining the primary pore space (Figure 2c) [13,25,29]. However, we found that if there are no secondary pores (mainly OM pores) after deep burial of 6000–7000 m [39], the preserved primary interP pores are very few in overmature shale samples (Figure 4). For example, although the rigid particle content of the Y101H4-4-7 sample is higher than Y101H2-7-1 and the clays content is not significantly different (Table 2), the OM content of Y101H4-4-7 is only 0.50% while that of Y101H2-7-1 is 3.39%. Regardless of the number of pores observed in SEM images (Figures 4 and 13) or the results of gas adsorption experiments (Table 3), the pore development of Y101H2-7-1 is much higher than that of Y101H4-4-7. Due to the strong heterogeneity of shale composition, rigid minerals and ductile ones are mixed together with varied particle size, which cannot form an effective support [63].

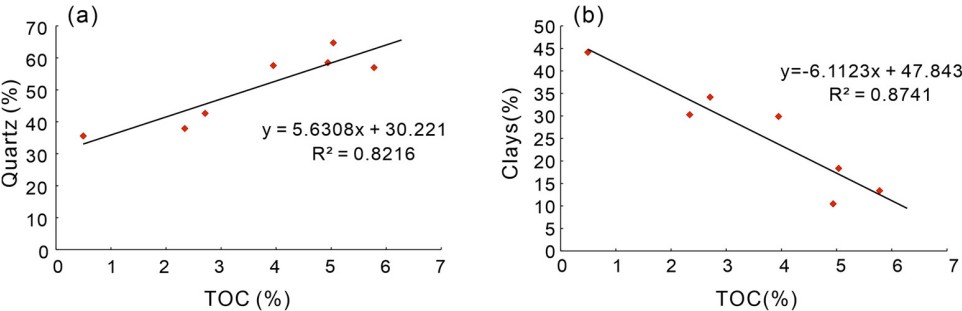

**Figure 10.** The relationship between the abundance of quartz, clays, and TOC of seven samples selected from well Y101H4-4. (**a**) the relationship between the content of quartz and TOC; (**b**) the relationship between the content of clays and TOC.

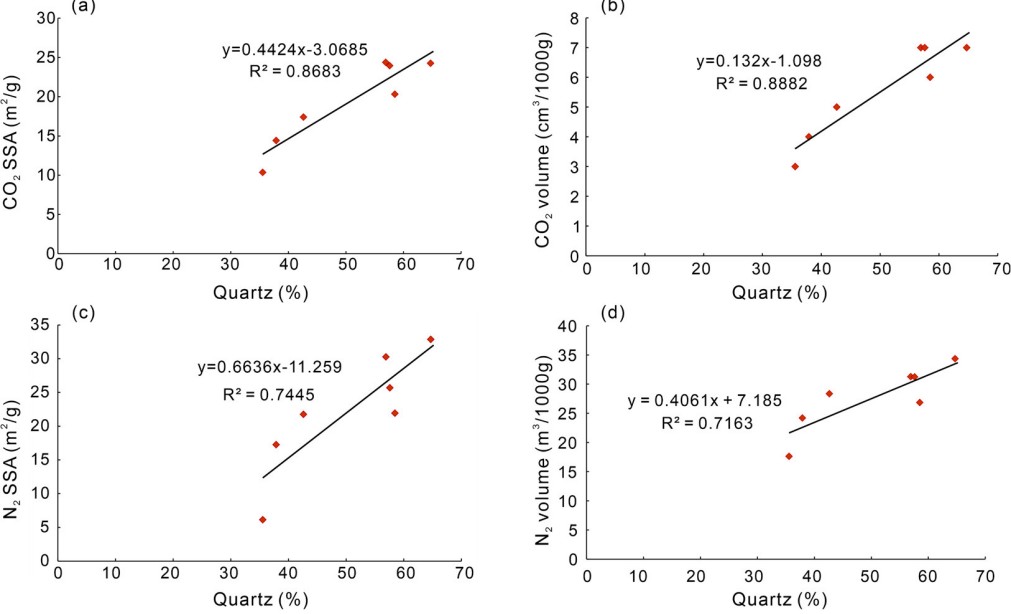

**Figure 11.** The relationship between the abundance of quartz and pore structure parameters of seven samples from well Y101H4-4. (**a**) the correlation between the quartz content and the value of $CO_2$ SSA; (**b**) the correlation between the quartz content and $CO_2$ volume; (**c**) the correlation between the quartz content and $N_2$ SSA; (**d**) the correlation between the quartz content and $N_2$ volume.

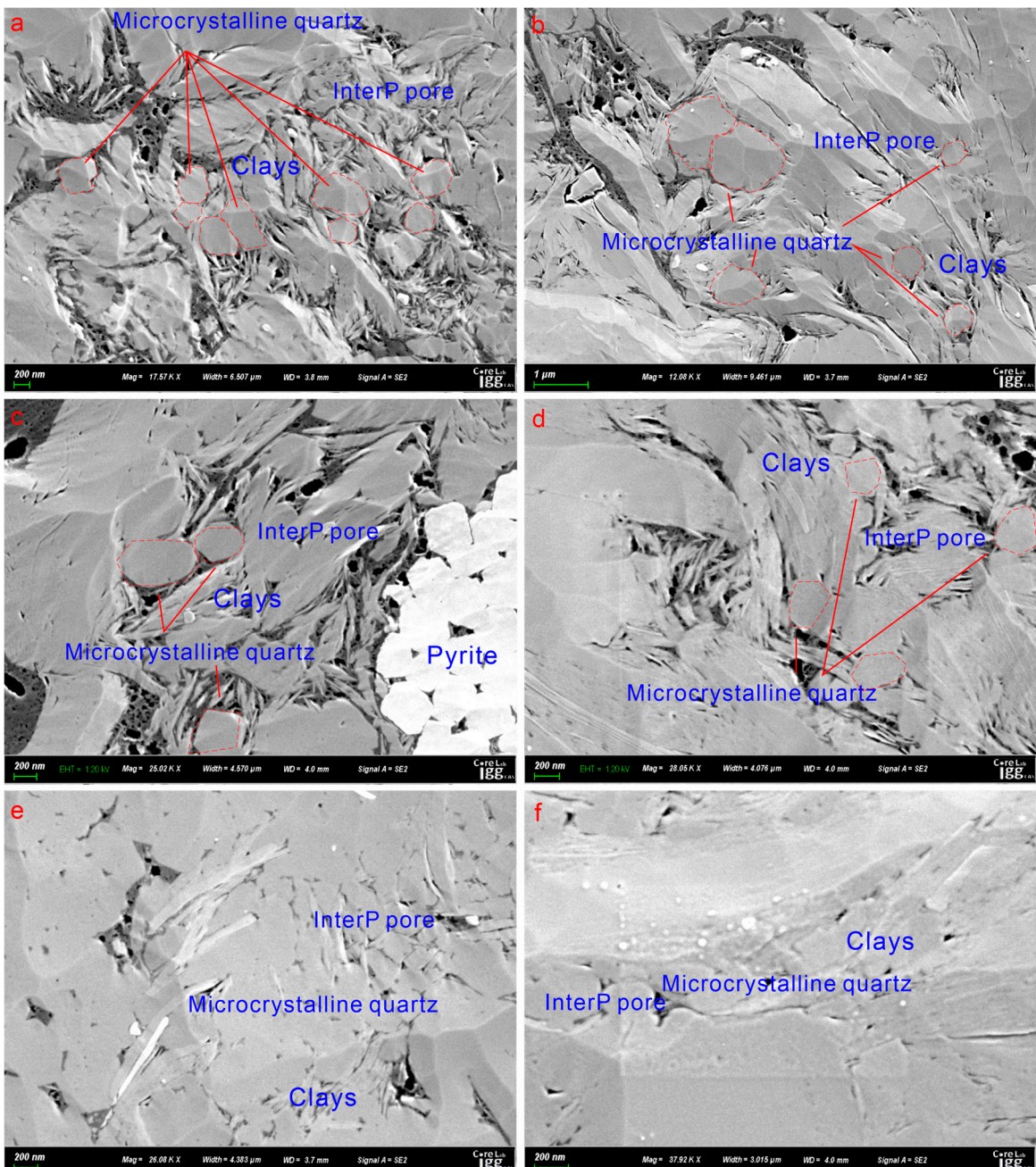

**Figure 12.** Quartz transformed from clay minerals and considerable pores developed during this process of illitization. (**a**,**b**) sample ID = Y101H4-4-1; (**c**,**d**) sample ID = Y101H4-4-2; (**e**) sample ID = L204-3; (**f**) sample ID = L205-1.

Carbonates, which are abundant in calcareous shales, have a dual effect on shale reservoirs. On one hand, carbonate cementation and precipitates destroy pores and reduce porosity during diagenesis [14,35,62]; on the other hand, they can improve the brittleness of shale reservoirs and play a beneficial role in the later hydraulic fracturing [64,65]. However, their chemical properties are unstable, making them easily dissolved by organic acids generated during the thermal maturation of sedimentary OM, resulting in the formation of dissolution pores [6,66]. Furthermore, the connectivity between pores within carbonates is limited (Figure 6), suggesting that not all pores within carbonates may be equally effective as shale gas storage space. Therefore, the final effect of carbonate minerals on the pore space of a shale reservoir depends on the relative degree of their destructive and constructive

effects. The samples in this paper show that the number of constructive pores is almost the same as destructive ones, and as a result, carbonate minerals do not contribute significantly to the total pores volume [14,35].

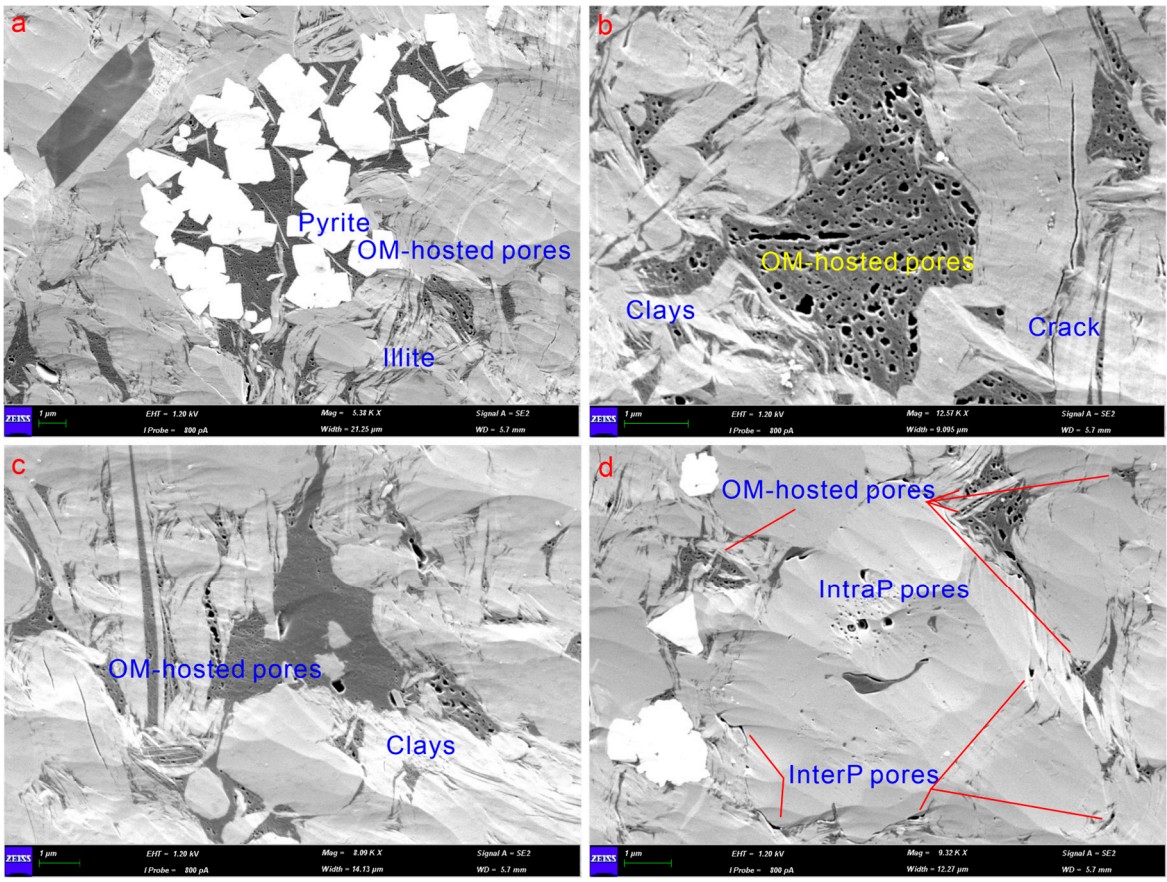

**Figure 13.** Pores developed in the sample Y101H2-7-1 with 3.39% TOC. (**a**) OM-hosted pores developed in the sample Y101H2-7-1; (**b**) OM-hosted pores and crack developed in the sample Y101H2-7-1; (**c**) OM-hosted pores developed between clays; (**d**) OM-hosted pores and interP pores developed in the sample Y101H2-7-1.

Feldspar, which has a relatively small content in shales, has the same impact on shale pores as carbonates [4,34,62]. Cementation decreases pores while dissolution increases pore space [4,57]. In general, it has no significant contribution to the total pore SSA and volume of shale reservoir due to its small content (Table 4).

Pyrite, though interP pores developed between pyrites or intraP pores developed within it (Figure 6a,b), is too small in content to have affected porosity significantly.

Although rigid particles can improve reservoir brittleness, making it more conducive to later hydraulic fracturing [1,64,65], they generally contribute little to total pore volume and SSA. Because of the different particle sizes, it is difficult to form effective support.

### 4.2. Contribution of Clays to Pore Development

Clays, the second-most abundant component in the Wufeng–Longmaxi Shale, have been demonstrated to be easily compacted [8,14,63]. The layered structure of clay contributes positively towards porosity [8]. However, if there is no force bearing inside or sheltering outside (Figure 14), these pore spaces may disappear due to compaction [8,67,68]. It has been reported that the porosity of original loose clays decreased by 85% after compaction [62]. Although it is also reported that rigid particles such as quartz, pyrite, and calcite can provide support to inhibit pore reduction [20], the heterogeneity of shale composition makes the pores between particles more easily filled rather than preserved (Figure 4).

In theory, rigid particles with similar size can resist compaction by point contact between rigid grains and form a stiff framework [54], thus preserving interP pore spaces. However, natural shales are a mixture of clays and other minerals that make it difficult to form an effective support and are more easily compacted due to differences in hardness and particle size [62]. Clays can adsorb OM on their surface and play a key role in the burial and preservation of OM [62,69,70].

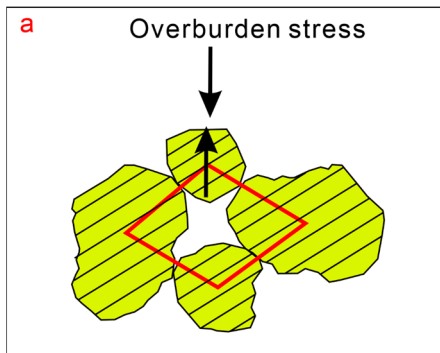 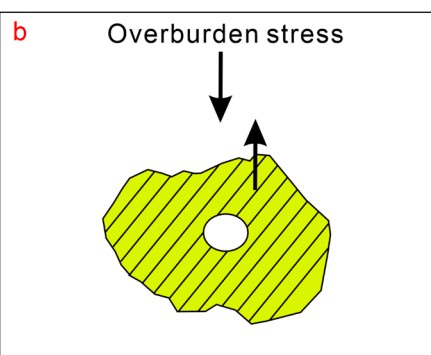 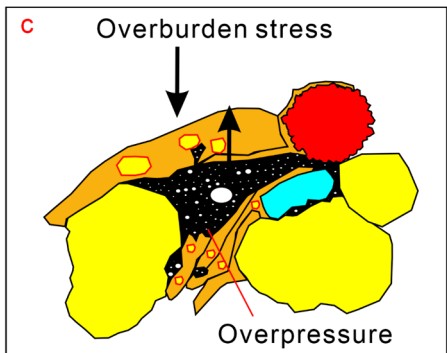

**Figure 14.** Pores preservation mechanism: (**a**) interP pore supported by rigid minerals, and the overburden stress of the overlying strata acting on the rigid framework; (**b**) intraP pore developed within the rigid minerals, and the overburden stress of the overlying strata acting on the rigid grain; (**c**) pores associated with plastic minerals such as OM or clays, and the pores supported by the overpressure inside.

Consequently, a significant amount of OM can be buried and deposited within clays. During the process of deep burial, typically above 70 °C, clays gradually transform from smectite to illite, leading to the release of significant amounts of silica that precipitates as microcrystalline quartz within the clays [24,54,62]. This process of illitization stiffens the shale framework and increases the brittleness of the reservoir [61], while also reducing the volume of clays and developing considerable pores (Figure 12) [24,54,67]. On the other hand, illitization is generally associated with hydrocarbon generation [71], as OM generates considerable amounts of hydrocarbons during thermal evolution, creating an overpressure environment in the reservoir [72,73]. Under the effect of pressure difference, the fluid may migrate into adjacent clays, becoming the force-bearing body in the pores (Figure 14c) and inhibiting pore collapse [74]. Therefore, we observe a large number of pores developing between the clays and rigid minerals in samples with relatively high TOC content (Figure 12). We infer that this phenomenon occurs because the fluid generated by hydrocarbon generation occupies this part of pore space in the clays, and the overpressure formed by the fluid counteracts the pressure of the overlying strata (Figure 14c), inhibiting the collapse of pores.

Overall, clay minerals can not only produce a large number of secondary pores during burial but can also be easily compacted due to their compressibility. Although some previous studies obtained a negative correlation based on the one-dimensional linear regression analysis of clay mineral content and pore parameters [14,34], in this study, the MLR models pointed out that clay mineral content significantly contributes to the total SSA and total pore volume of shale pores. In fact, simple linear regression analysis ignores the collinearity between multiple factors. For example, clay minerals are negatively correlated with TOC (Figure 10b), which may be the reason why clay minerals are negatively correlated with pore structure parameters. It should be pointed out that clays have no contribution to the maintenance of primary pores due to their ductile nature, but secondary pores may be generated during diagenesis and can be preserved if hydrocarbon generation generates overpressure to provide support (Figure 14c).

### 4.3. Contribution of OM to Pore Development

OM contributes significantly to the shale reservoir space, particularly in overmature marine shales [20]. However, the presence of OM-hosted pores depends strongly on hydrocarbon generation and thermal evolution, making them the primary pore types in overmature shales [18,20,68,72]. The OM was deposited concurrently with minerals and served as part of the sediments (Figure 15a). As burial depth increased and stress acted on the sediment, ductile minerals such as OM and clays deformed under the effect of in situ overburden stress and occupied the surrounding interP pores (Figure 15b) [68,74]. Meanwhile, as thermal maturity increased, OM evolved further, generating hydrocarbons such as bitumen, oil, gas, and so forth [72]. During this thermal evolution of OM, OM-hosted pores developed in residual OM due to the thermal cracking of in situ OM (Figure 15b) [31,68,74], while overpressure built up due to hydrocarbon generation [72,73]. Driven by the overpressure, parts of the mobile products, including liquid bitumen and oil, migrated to adjacent pores (Figure 15c), including interP pores supported by rigid minerals and pores within clays.

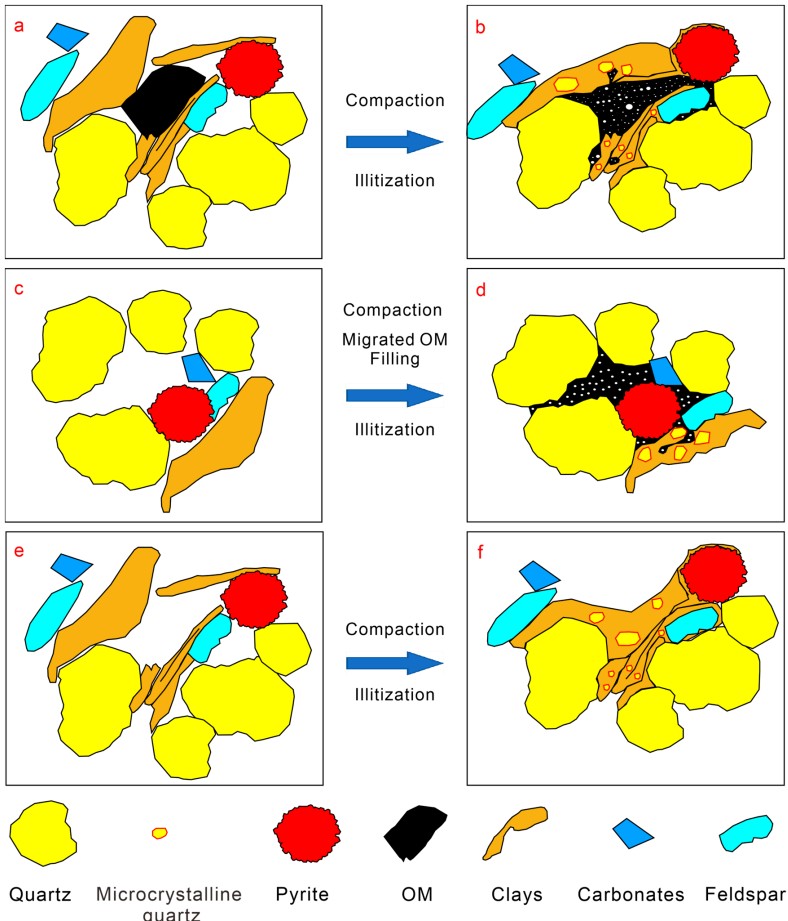

**Figure 15.** Mechanism of OM on pore preservation: (**a**) OM in situ deposited as part of the sediment framework; (**b**) ductile minerals, such as clays and OM, changed during compaction, secondary pores and microcrystalline quartz developed during diagenesis; (**c**) minerals deposited without OM; (**d**) interP pores associated with OM filling, and secondary pores developed during diagenesis and preserved by OM; (**e**) minerals deposited with higher clays but no OM; (**f**) clays compacted during compaction with pore destruction.

The filling of migrated OM and subsequent products of the further thermal evolution of the OM in these pores prevented the pores from collapsing due to further compaction (Figure 15d). For overmature shales, both residual OM and migrated OM would be further cracked with increasing thermal maturity, and various scales of pores developed within

residual OM and migrated OM (Figure 15b,d), while the high cementation of surrounding mineral would form a relatively closed system, thus forming overpressure to preserve pores from compaction. At the same time, illitization of clays generated pores within clays (Figure 15b,d,f). If the sediments were deposited without OM and no OM migrated during burial (Figure 15e), pores would strongly reduce due to compaction (Figures 4 and 15f), but sediments with OM is another case. For both the OM attached to the surface of clays during deposition and the OM that migrated to the clays during burial, after thermal decomposition, its products, which can act as a bearer to inhibit the collapse of pores (Figure 14c), filled the pores generated by the diagenesis of clays (Figure 15b,d). In other words, pores not only can be preserved by rigid minerals (Figure 14a,b) [60] but also by products of hydrocarbon pyrolysis, such as oil and gas, especially under overpressure (Figure 14c). In deep burial shale, OM not only provides pore space to store shale gas, but its products also support pores.

## 5. Conclusions

The contribution to pore structure of shale components were analyzed by MLR to determine the key factor controlling the gas capacities of marine shales. Our investigation poses a challenge to the previous understanding. Through detailed study on the contribution of shale composition to the pores in Wufeng–Longmaxi Shale, important new information is provided for further understanding of the nature of shale pore systems. Conclusions are as follows: (1) The Wufeng–Longmaxi Shale is dominated by quartz, clays, carbonates, feldspar, and OM, according to XRD analysis. (2) OM and clays are the primary contributors to the storage space in overmature shale, indicated by the MLR, and OM contributes more to the pore structure. Indeed, OM not only provides pore space to store shale gas, but its products also support pores, including OM-hosted pores and pores generated by clay transformation. (3) Total SSA and volume of shale pores increases with the contents of TOC and clays. (4) Quartz, the most abundant component in shale, hardly contributes to the SSA and volume of shale pores. The quartz content of some samples has a positive correlation with the SSA and volume of shale pores because this part of quartz is biogenic, and its abundance is positively related to TOC. Thus, there is a collinear relationship between quartz and TOC. The increment of the pore SSA and pore volume of this part comes from OM rather than quartz. (5) Although carbonates and feldspar are prone to produce dissolution pores during diagenesis, their amount contributes little to the total pore volume and SSA of shale. (6) Shale pores can be preserved in two ways: overpressure existing inside or shelter existing outside the pores.

We acknowledge that we neglected the differences of the thermal maturity and its influence on pore structure. The result of this study explains the contribution of the composition of overmature shale to the SSA and volume of pores. As the experiment was only carried out by gas adsorption without mercury injection capillary pressure or other methods, it is still unclear how shale composition affects the pore structure of macropores with width over 300 nm. A total of 29 samples were analyzed for statistical analysis. However, being a statistical analysis, the correlation models need more data to be more convincing. We hope this paper will be conducive to understanding the mechanism of pore development in overmature shales.

**Author Contributions:** Conceptualization, L.X. and R.P.; methodology, H.H.; software, J.M.; validation, H.H.; formal analysis, L.X.; investigation, H.H.; resources and data curation, L.X.; writing—original draft preparation, L.X.; writing—review and editing, R.P.; supervision, J.M.; project administration, H.H.; funding acquisition, R.P. All authors have read and agreed to the published version of the manuscript.

**Funding:** This research was funded by the National Natural Science Foundation of China (grant number 41472123).

**Institutional Review Board Statement:** Not applicable.

**Informed Consent Statement:** Not applicable.

**Data Availability Statement:** Not applicable.

**Acknowledgments:** The authors would like to thank PetroChina Southwest Oil & Gas field Company for providing core samples. Many thanks also go to the anonymous reviewers for their insightful scientific comments and suggestions that significantly enhanced the quality of the original manuscript. Gratitude should go to Academic Editor Anabela Oliveira and Assistant Editor Sumesa Puangpee for their valuable comments and language improvements that greatly improved the original manuscript.

**Conflicts of Interest:** The authors declare no conflict of interest.

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
