# Peer review of "Contribution of Various Shale Components to Pore System: Insights from Attributes Analysis"

_jmse, doi:10.3390/jmse11071327_

Round 1
Reviewer 1 Report
This is a very interesting study investigating any correlation (or lack of) between porosity in Wufeng-Longmaxi shale with its TOC, quartz, carbonate and clay mineral content. This is important to characterise the storage capacity of these subsurface reservoirs. Different methods including high resolution imaging, compositional analyses and a robust statistical approach was applied for the study.
The study demonstrates, in agreement with previous studies, that the porosity of the shale was mainly controlled by the TOC. Further, this study also suggests that because biogenic quartz is correlated to TOC, the quartz is also correlated with porosity but is not the determining factor. While the porosity in carbonate and feldspar are insignificant, for clay minerals the porosity might be important if the pores are preserved due to overpressure.
Further feedback is provided in the attached file.

Suggestions in the attached file.
Reviewer 2 Report
The paper is well written and easy to comprehend. The authors has put forth the entire methodology to at most scientific way possible. However, if the authors could add a section where the limitations of the study could be mentioned. Future research direction can also be pointed out. Additionally do the authors think that age of formation can affect its porosity? How can this study be related to similar formation across the globe?
Reviewer 3 Report
The Paper is good but needs some updates for references see the attached pdf

Reviewer 4 Report
The paper, presents presents interesting ideas, on a matter of clear scientific and industrial interest.
The main contribution of the paper is the analysis of pore shell. Results obtained are complete, well presented and cleary explained.
Introduction and conlusion part can be impreoved.
So, I recommend that this paper be accepted after minor revision:
1. “Intoduction” paragraph can be improved, well explaininhg the aim of your research and novelty linked to your work
2. Conclusion paragraph must be enriched by addressing the conclusion the consequences of the results on future progress and perspectives
3. English language have to be improoved.
The english language is quite poor, and in some passages, it doesn't sound "scientific".
You should enrich it and make it more fluent.
Reviewer 5 Report
Good and valuable research has been done.
This study missions that, Shale pore systems are the result of the geological evolution of different matrix assemblages, and the composition of gas shale is considered to affect the pore systems in shale reservoirs. This study aimed to investigate the impact of both organic and inorganic constituents on the shale pore system, including specific surface area (SSA) and pore volume in Wufeng-Longmaxi Shale. Multiple linear regression (MLR) was employed to examine the contributions of different components to shale pore structure. The pore structure parameters, including pore SSA and pore volume, were obtained by gas adsorption experiments in 32 Wufeng-Longmaxi Shale (Late Ordovician-Early Silurian) samples. Both pore SSA and pore volume were calculated by the density functional theory (DFT) model on shale samples, and the pore types were determined by high-resolution field emission scanning electron microscopy (FE-SEM). The results of the X-ray diffractometer (XRD) analysis indicate that the Wufeng-Longmaxi Shale is dominated by quartz, clays, carbonates, feldspar, pyrite, and organic matter. Four models were made using SPSS software, all of which showed significant correlation between shale pore size and organic matter (OM) and clays. The content of organicmatter played the biggest role in determining the size and structure of the pores. Although the content of quartz is the most and serves as a rigid skeleton in shale reservoirs, it has complicated effects on the pore structure. In this study, most of the quartz is biogenetic and part of it is transformed from clays in deep shale. Therefore, these two parts of quartz are respectively related to organic matter and clays. In essence, the pores related to these two parts of quartz should be attributed to organic matter and clays, which also support the conclusion of the MLR models. evolution of bonding performance of HIRA(High Intensity and Rapid Agent) anchor solids with maintenance time, the evolution characteristics of bond strength and stress distribution at the interface between HIRA-based anchor solids and geotechnical body under different maintenance time and the fine damage pattern of anchor solids were studied by indoor pull-out test of anchor solids, and the comparative analysis was performed with 42.5 grade ordinary Portland cement(hereinafter referred to as P.O 42.5). The results show as follows. The early strength and rapid setting characteristics of HIRA type material are obvious, and the difference between its average peak bond strength and that of cement is 10.45 times; The shear stress distribution has obvious stress concentration characteristics, and the peak value will appear and shift with the increase of load, and the peak shift of HIRA anchor solid occurs earlier than that of cement. Due to different stress levels, the damage of HIRA anchor solid after pulled out increases with the increase of maintenance time, while that of cement gradually becomes more severe, the overall damage of HIRA material is generally lower than that of cement in the same period.
The manuscript is well organized and has good content.
In the opinion of the reviewer, this manuscript note could be accepted after the major corrections and re-evaluation.
1- Authors are recommended to discuss the obtained results with literature in more detail.
2- Authors are recommended to emphasis the novelty and significance of the study in more detail.
3- Please use the appropriate font color in Figures 2-7. The color of some tests in the print is faint.4- Authors are recommended to provide a more complete explanation in the “Contribution of OM to pore development” section.
5- It is recommended to prepare a table in the introduction section and list the works done in order of year.
6- Some of the references provided are old. It is recommended that a number of recent papers that are new and have been published in the last five years be used in the introduction and references list.
Round 2
Reviewer 5 Report
The desired corrections have been made. In my opinion, the article can be accepted.